Forbidden links, trait matching and modularity in plant-hummingbird networks: Are specialized modules characterized by higher phenotypic floral integration?

Izquierdo-Palma Jaume 1
http://orcid.org/0000-0003-4838-5432 Arizmendi Maria del Coro 1 coro@unam.mx
http://orcid.org/0000-0001-6037-0327 Lara Carlos 2
http://orcid.org/0000-0002-1124-1163 Ornelas Juan Francisco 3
1 Laboratorio de Ecología, UBIPRO, Facultad de Estudios Superiores Iztacala, Universidad Nacional Autónoma de México , Tlalnepantla de Baz, Estado de México , Mexico
2 Centro de Investigación en Ciencias Biológicas, Universidad Autónoma de Tlaxcala , San Felipe Ixtacuixtla, Tlaxcala , Mexico
3 Departamento de Biología Evolutiva, Instituto de Ecología A.C. , Xalapa, Veracruz , Mexico
Pimm Stuart
Electronic publication date: 2021 Mar 10
Publication date: 2021
Volume: 9
Electronic Location ID: e10974
Received 2020 Jul 15; Accepted 2021 Jan 29
Copyright: © 2021 Izquierdo-Palma et al.
Copyright year: 2021
Copyright holder: Izquierdo-Palma et al.
License: This is an open access article distributed under the terms of the Creative Commons Attribution License, which permits unrestricted use, distribution, reproduction and adaptation in any medium and for any purpose provided that it is properly attributed. For attribution, the original author(s), title, publication source (PeerJ) and either DOI or URL of the article must be cited.
License URL: https://creativecommons.org/licenses/by/4.0/

Keywords: Interaction networks, Hummingbirds, Biotic interactions, Pollination, Mexico, Forbidden links, Biotic specialization, Modularity, Phenotypic floral integration, Neotropical rainforest

Funding: Universidad Nacional Autónoma de México (UNAM) PAPIIT Research Funds IN216617 Consejo Nacional de Ciencia y Tecnología (CONACyT) 855974 This work was supported by Universidad Nacional Autónoma de México (UNAM) PAPIIT research funds IN216617 and Consejo Nacional de Ciencia y Tecnología (CONACyT) with the JI-P doctoral scholarship (No. 855974). The funders had no role in study design, data collection and analysis, decision to publish, or preparation of the manuscript.

==============================
Background

Plant-pollinator mutualistic networks show non-random structural properties that promote species coexistence. However, these networks show high variability in the interacting species and their connections. Mismatch between plant and pollinator attributes can prevent interactions, while trait matching can enable exclusive access, promoting pollinators’ niche partitioning and, ultimately, modularity. Thus, plants belonging to specialized modules should integrate their floral traits to optimize the pollination function. Herein, we aimed to analyze the biological processes involved in the structuring of plant-hummingbird networks by linking network morphological constraints, specialization, modularity and phenotypic floral integration.

Methods

We investigated the understory plant-hummingbird network of two adjacent habitats in the Lacandona rainforest of Mexico, one characterized by lowland rainforest and the other by savanna-like vegetation. We performed monthly censuses to record plant-hummingbird interactions for 2 years (2018–2020). We also took hummingbird bill measurements and floral and nectar measurements. We summarized the interactions in a bipartite matrix and estimated three network descriptors: connectance, complementary specialization (H2’), and nestedness. We also analyzed the modularity and average phenotypic floral integration index of each module.

Results

Both habitats showed strong differences in the plant assemblage and network dynamics but were interconnected by the same four hummingbird species, two Hermits and two Emeralds, forming a single network of interaction. The whole network showed low levels of connectance (0.35) and high specialization (H2’ = 0.87). Flower morphologies ranged from generalized to specialized, but trait matching was an important network structurer. Modularity was associated with morphological specialization. The Hermits Phaethornis longirostris and P. striigularis each formed a module by themselves, and a third module was formed by the less-specialized Emeralds: Chlorestes candida and Amazilia tzacatl. The floral integration values were higher in specialized modules but not significantly higher than that formed by generalist species.

Conclusions

Our findings suggest that biological processes derived from both trait matching and “forbidden” links, or nonmatched morphological attributes, might be important network drivers in tropical plant-hummingbird systems while morphological specialization plays a minor role in the phenotypic floral integration. The broad variety of corolla and bill shapes promoted niche partitioning, resulting in the modular organization of the assemblage according to morphological specialization. However, more research adding larger datasets of both the number of modules and pollination networks for a wider region is needed to conclude whether phenotypic floral integration increases with morphological specialization in plant-hummingbird systems.

Introduction

Plant and animal species are integrated in complex networks of interdependencies forming ecological communities playing an important role in the generation of Earth’s biodiversity (Bascompte & Jordano, 2007). Plant-pollinator mutualistic networks are linked by trophic interactions where plants act as primary producers and animals as a special subset of primary consumers that feed on nectar and/or pollen (Bascompte & Jordano, 2007; Waser et al., 1996). The strength of their interactions varies, resulting in large, complex networks in which interacting species impose reciprocal selective pressures as they interrelate over ecological and evolutionary time (Bascompte & Jordano, 2007; Thomson & Wilson, 2008; Waser et al., 1996). Over the last decade, the study of mutualistic networks has changed radically, along with the theory (Bascompte & Jordano, 2007; Blüthgen et al., 2007; Olesen et al., 2007; Ings et al., 2009; Vázquez et al., 2009; Dalsgaard et al., 2011), and new powerful analytical tools have been proposed (Dormann, Gruber & Fründ, 2008; Dormann & Strauss, 2014). These advances have shown that non-random structural properties can be characterized by certain network metrics. For example, most interaction networks show a nested structure (i.e., specialists interact with subsets of species with which generalists also interact) and varying levels of connectivity among species. Both properties facilitate species coexistence by minimizing competition relative to facilitation, supporting greater biodiversity (Bascompte et al., 2003; Olesen et al., 2006; Verdú & Valiente-Banuet, 2008; Bastolla et al., 2009; Sugihara & Ye, 2009). Furthermore, most ecological networks are strongly asymmetric (i.e., a plant species might heavily depend on a pollen-vector species that, in turn, is only weakly dependent on that plant species). Thus, the community is structured around a central core of generalists, offering robustness and resilience to the random loss of species (Vázquez & Aizen, 2003; Bascompte, Jordano & Olesen, 2006; Guimarães et al., 2006; Blüthgen et al., 2007; Bascompte & Jordano, 2013).

Although the discussion on the architecture of mutualistic networks is quite settled, the underlying structuring mechanisms are still being debated (Maruyama et al., 2014; Vizentin-Bugoni, Maruyama & Sazima, 2014; Araujo et al., 2018). Mutualistic networks tend to be very heterogeneous in the number of interacting species at each level and the distribution of their connections, such as, for example, between animal-dispersed fruits and their dispersers or animal-pollinated angiosperms and their pollinators (Bascompte & Jordano, 2007). Two major biological mechanisms that influence the structure of networks are the “complementary traits” and “barrier traits” (Santamaría & Rodríguez-Gironés, 2007). For complementary traits, the interaction is determined by the similarity between the reward that the plant has to offer and the resource that the pollinator seeks. This mechanism can progressively generate co-specialization. For example, flowers that are mainly pollinated by birds are red in color, matching the perceptual system of their avian pollinators who show a preference for this color when looking for nectar resources (Niovi Jones & Reithel, 2001). For the mechanism of barrier traits, the interaction is related to the ability of the pollinator to reach the reward offered by the flower. Only those pollinators whose traits allow them to overcome the floral barriers are able to access the reward. For example, Hermit hummingbirds with typically long and curved bills have access to flowers with long and curved corolla tubes that hummingbird species with short and straight bills cannot access (Maglianesi et al., 2014). Consequently, competition for shared floral resources is reduced (Feinsinger, 1976; Ings et al., 2009; McGuire et al., 2009; Abrahamczyk & Kessler, 2010; Maglianesi et al., 2014). These exploitation barriers imposed by the mismatch of biological attributes that lead to a decrease in the connectivity and/or strength of the interactions in ecological networks can be called “forbidden links” (Santamaría & Rodríguez-Gironés, 2007; Olesen et al., 2011).

Species abundances can be as important or even more important as species traits in structuring the ecological interaction networks of local communities (e.g., common in insect-pollination networks; Vázquez, Chacoff & Cagnolo, 2009). However, plant-hummingbird mutualistic networks are considered a specialized system, particularly those located near the Equator due to higher productivity and the relatively stable and predictable availability of resources throughout the year (Feinsinger & Colwell, 1978; Dalsgaard et al., 2011; Belmaker, Sekercioglu & Jetz, 2012; Zanata et al., 2017). Hence, several studies have shown that mismatches in species morphology and phenology play a major role in structuring interactions in plant-hummingbird systems (Stiles, 1975, 1978; Lara, 2006; Maglianesi et al., 2014; Maruyama et al., 2014; Vizentin-Bugoni, Maruyama & Sazima, 2014; Sonne et al., 2019). Hummingbird clades have characteristic morphologies that influence resource use, flight capabilities, competitive skills and environmental filtering, important mechanisms structuring hummingbird communities (Rodríguez-Flores et al., 2019). Furthermore, hummingbird morphological traits, such as bill length and curvature and body mass, have also been hypothesized to play a role in the specialization of hummingbird interactions. Meanwhile, corolla length and curvature and nectar volume are floral traits associated with the specialization of hummingbird-pollinated plants (Maruyama et al., 2014, 2018; Maglianesi et al., 2014, 2015; Dehling et al., 2016). Thus, specialization in trophic resources’ use by hummingbird assemblages, which vary in its type and/or its strength, can arise in niche segregation of the community due to the differential access to flowers (Feinsinger & Colwell, 1978).

The extent to which connectivity is limited by both morphological mismatches and spatio-temporal constraints often results in resource segregation, leading to niche partitioning and, consequently, a modular structure in ecological networks. Network modularity reflects the tendency of a set of species to interact predominantly with species within the set and less frequently with species in other sets. Modularity implies that species can be grouped (i.e., modules) in such a way that weakly interlinked subsets of species are strongly connected internally (Olesen et al., 2007). Modules can provide information on the dynamics of ecological communities by identifying specialized functional groups of pollinators and floral traits (Newman, 2006; Olesen et al., 2007; Danieli-Silva et al., 2012; Dormann & Strauss, 2014; Maruyama et al., 2014). Some studies support the idea that modularity increases with the increased specialization of the interacting species (Lewinsohn et al., 2006; Guimarães et al., 2006) and is positively related with network size (Olesen et al., 2007). Thus, in little-connected and highly nested networks, which are common in the tropics (Olesen & Jordano, 2002), there is a higher probability of modularity (Fortuna et al., 2010).

Specialized modules drive strong plant-pollinator relationships, leading to the diffuse co-evolution of complementary morphological traits (Sazatornil et al., 2016). Furthermore, selection imposed by pollinators can be an important mechanism shaping the patterns of variation and covariation of floral traits, particularly in plant species with specialized pollination. Such is the case in the Phaethornis-Heliconia pollination system (Armbruster, 1991; Fenster, 1991; Herrera et al., 2002; Pérez, Arroyo & Medel, 2007; Ordano et al., 2008). In this context, flowers can be seen as suites or units of traits that require a precise configuration and arrangement of their sexual organs for proper pollination (Bissell & Diggle, 2008). From this multi-trait view, some authors have argued that selection imposed by specialized pollinators reduces phenotypic variability and favors the integration of subsets of floral traits (Pleiades), that is, correlations among traits within functional units usually involved in one ecological function (Berg, 1960; Stebbins, 1970; Conner & Via, 1993; Rosas-Guerrero et al., 2011). However, other factors may be linked with phenotypic floral integration, such as the breeding system (Anderson & Busch, 2006; Rosas-Guerrero et al., 2011) and developmental-genetic factors (Conner, 2002; Smith & Rausher, 2008). Nonetheless, the relationship between specialization and niche partitioning in hummingbird assemblages and its possible effects on the phenotypic floral integration of interacting plant species are poorly understood (Berg, 1960; Stebbins, 1970; Conner & Via, 1993; Rosas-Guerrero et al., 2011).

In this study, we assessed the association among biological processes involved in the structuring of plant-hummingbird networks, niche segregation by ecological specialization, and phenotypic floral integration of the plant assemblages. We used the phenotypic floral integration index as a measurable estimate of the magnitude and pattern of covariation among sets of functionally related floral traits to obtain new insights into the link between modularity and specialization in plant-hummingbird mutualistic networks (Ordano et al., 2008; Rosas-Guerrero et al., 2011; Dormann & Strauss, 2014). To achieve this, we investigated the network architecture by descriptors commonly used in similar studies and identified the possible underlying biological processes, that is, trait matching and forbidden links. Then, we analyzed the modularity and floral integration index of each module to identify patterns of covariation. In particular, we examined the understory plant-hummingbird mutualistic networks in two distinct adjacent habitats in Mexico, addressing the following questions: (1) Are there differences between the two habitats in the composition of interacting species and their network metrics? (2) What is the main network structuring each habitat? And, (3) is there a relationship between module specialization and phenotypic floral integration? To our knowledge, the linking of network morphological constraints, specialization, modularity and phenotypic floral integration is a new approach in the study of plant-hummingbird interaction networks.

Materials & methods

Study area

Fieldwork was carried out at the Chajul Biological Station located in the Montes Azules Biosphere Reserve (16°06′ N; 90°56′ E) within the Lacandon region in southern Mexico a few kilometers from the Guatemalan border. The study area covers an extension of ~331,200 ha and is situated from 150 to 1,500 m above sea level (m.a.s.l.). Mean rainfall in Chajul is around 3,000 mm, of which ca. 70% is concentrated from June to October. The short dry season occurs between February and April. The mean annual temperature is 22.5 °C (Siebe et al., 1996; Carabias, De la Maza & Cadena, 2015). The dominant vegetation around the field station is lowland evergreen tropical rainforest (hereafter “rainforest” habitat) with some variability influenced by the soil properties and proximity to water bodies. Near streams and rivers, there is riparian vegetation and sections of flooded plains. In areas surrounding the field station where the anthropic impact has been more intense in recent times, the vegetation mainly consists of secondary forest and abandoned fields in different stages of ecological succession. There is also some hilly terrain (highest elevation: 230 m.a.s.l) with thin and poor soils, and the vegetation here is savanna-like with low and scattered trees and an understory characterized by abundant grass (Scleria melaleuca; hereafter “savanna” habitat) (Miranda & Hernández, 1963; Rzedowski & Huerta, 1994; Siebe et al., 1996). We collected data from January 2018 to January 2020 along trails 6,700 m long in both study habitats.

Phenology of hummingbirds and their plants

At monthly intervals, we recorded the hummingbird species and their numbers along six study trails in both habitats. Walking censuses began around 7:00 AM and ended at 1:30 PM. All hummingbird-pollinated plant species (individuals or floral patches) flowering within 2.5 m on each side of the trails were counted at a maximum height of 5 m. We focused on the understory plant community for logistical reasons. Flowering plants in the canopy are difficult to see from the ground, which may result in an underestimation of the number of plant individuals and interactions with their pollinators. Binoculars (Nikon 10 × 42) and a field guide were used to identify hummingbirds (Arizmendi & Berlanga, 2014), and plant specimens were identified at the Chajul Biological Station herbarium. The scientific names of plants were validated using “The Plant List” online database (http://www.theplantlist.org/) and hummingbird scientific names using the IOC World Bird List (www.worldbirdnames.org).

Floral measures and bill morphology

Because hummingbirds commonly visited the flowers of the plant species we observed during the censuses, we quantified several floral traits presumed to be associated with hummingbird pollinator attraction and pollen transfer efficiency (Wolf, Stiles & Hainsworth, 1976; Stiles, 1995). The flower morphology was characterized by measuring the corolla length and curvature, as these are the primary constraints determining the ability of hummingbirds to reach the nectar. We measured the effective corolla length (i.e., distance from the nectary to the distal portion of the flower, which determines how far the bill of the feeding bird fits into the flower) (Wolf, Stiles & Hainsworth, 1976). In addition, we calculated the flower curvature following similar methodology used in hummingbird bills (Stiles, 1995; Rico-Guevara & Araya-Salas, 2015). We calculated a corolla curvature index as the arc:chord ratio of the corolla. Arc length was measured following the dorsal profile of the corolla from the calyx to the corolla tip, and the chord was measured as the straight-line distance from the calyx to the corolla tip. These measures were taken from lateral photographs using the ImageJ software (Schneider, Rasband & Eliceiri, 2012). Because the placement of pollen on the vector and its subsequent reception on the stigma is crucial to plant fitness, we measured the average stamen length and style length. For each plant species recorded in both habitats, we measured from 2 to 180 flowers collected from at least two individuals. Accumulated nectar was also quantified to determine reward availability for hummingbirds. From each plant species, 2–30 buds about to open were selected and placed in mesh bags (1-mm bridal tulle) to exclude hummingbird visitors and allow nectar to accumulate. After the flowers opened, the accumulated nectar was extracted, and the nectar was removed and measured after 24 h of nectar accumulation using calibrated micropipettes (5 μL) and a digital caliper (error: 0.1 mm). The sugar concentration (percentage sucrose) was measured by a hand-held pocket refractometer (range concentration 0–32° Brix units (°Bx); Atago, Tokyo, Japan). To characterize the main floral types of the whole plant assemblage, we performed a principal components analysis (PCA) of the measured floral traits (morphology and nectar) after discarding highly correlated variables through a correlation analysis.

The hummingbirds’ bill morphology was measured as the length of the exposed culmen and its curvature from voucher specimens housed at the collection of the Museo de Zoología, Facultad de Ciencias, Universidad Nacional Autónoma de México (MZFC, UNAM) (Phaethornis longirostris (n = 30), P. striigularis (n = 20), Amazilia tzacatl (n = 30), and Chlorestes candida (n = 28)). As hummingbirds can project their tongues to drink nectar (Paton & Collins, 1989), bill measurements that ignore tongue extension can underestimate birds’ capacity to access nectar. Because precise measurements of tongue length are unavailable for different hummingbird species, we added one-third to the bill length for each species (Vizentin-Bugoni, Maruyama & Sazima, 2014). To examine differences in bill shape, we calculated a bill curvature index as the arc:chord ratio of the exposed culmen (maxillary curvature) (Stiles, 1995; Rico-Guevara & Araya-Salas, 2015). Arc length was measured following the dorsal profile of the bill from the feathered base to the tip, and the chord was measured as the straight-line distance from the feathered base to the tip (Phaethornis longirostris (n = 18), P. striigularis (n = 14), Amazilia tzacatl (n = 13), and Chlorestes candida (n = 20)). These measures were taken from lateral photographs of the complete bill using the ImageJ software (Schneider, Rasband & Eliceiri, 2012). Furthermore, we obtained the average measures of total body weight to relate them with agonistic behavior and nutritional requirements (Phaethornis longirostris (n = 15), P. striigularis (n = 11), Amazilia tzacatl (n = 10) and Chlorestes candida (n = 12)). To obtain a closer relationship between the trait match of plant species and their hummingbird visitors, we calculated the average of each floral trait of the flowering plants visited by the hummingbird species. Then, comparisons were made with the average bill length and curvature.

Plant-hummingbird interactions

We built the plant-hummingbird mutualistic networks from a plant-centered approach (Jordano, 1987; Bosch et al., 2009). We did not record other pollinator interactions despite have been reported that many plants visited by hummingbirds are also visited by insects (Dalsgaard et al., 2009). Nevertheless, this fact should not affect to test our hypotheses due to the high specialization of our study system. Plant-insect pollination studies require another methodology with closer observations and adapted to the daily foraging activity of each invertebrate species. Because our study was based on mutualistic relationships, we only considered hummingbirds as potential pollinators. We recorded legitimate hummingbird visits, that is, when hummingbirds contacted the reproductive structures of the flowers. Each visit was defined as the moment a hummingbird probed one flower until it left the flowering plant/patch. We conducted from 8 to 50 h focal observations of each plant species (Vizentin-Bugoni et al., 2016). Most observations were conducted by video recording (GoPro Hero5), but in some cases (i.e., large floral patches or epiphytes with difficult access), we used binoculars (Nikon 10 × 42) to prevent underestimating the interactions. The observations were conducted from 07:00 AM to 11:30 AM, the time period of maximum foraging activity based on preliminary observations. Whenever possible, we conducted the observations at different plant individuals and locations to capture maximum variability. We observed a total of 657 h of plant-hummingbird interactions.

Analysis of interaction networks

We summarized the plant-hummingbird interactions in a bipartite matrix with each cell indicating the frequency of interactions. Because the two habitats are adjacent, they can form a single network. For this reason, we built a single interaction network for both habitats. However, to have a glimpse of the possible underlying biological processes modeling interactions in each habitat, we also built two separate interaction networks corresponding with the rainforest and savanna communities. Using these matrices, we estimated several network metrics of structure and specialization, which are detailed at following: (1) Connectance was calculated as the proportion of possible links in the network that are actually realized. If nonmatching species traits can prevent the occurrence of certain interactions (forbidden links), connectance is an estimate of how interactions are constrained in the communities. (2) Complementary specialization (H2’) estimates the exclusiveness of interactions considering the ecological specialization of a species (i.e., how connected a species is) and how these interactions differ among species. The H2’ index is useful for comparing ecological networks, as it is less affected by community size or sampling intensity (Blüthgen et al., 2007). (3) Nestedness was calculated using the ANINHADO software (Guimarães & Guimarães, 2006). We used two estimators, the NODF index, which uses qualitative presence/absence data and wNODF, which considers quantitative interaction data (Almeida-Neto et al., 2008; Almeida-Neto & Ulrich, 2011). (4) Modularity (Q), as defined above, was estimated for both quantitative and qualitative matrices. For the quantitative matrices, we used the QuaBiMo optimization algorithm (Dormann & Strauss, 2014). As the QuaBiMo algorithm has an iterative searching algorithm (values can slightly differ between runs), we chose the highest values from 10 independent runs. The modularity of the qualitative matrix was estimated in MODULAR (Marquitti et al., 2014), a stochastic algorithm, using Barber’s metric for bipartite networks (Barber, 2007) and following the recommended program settings (Marquitti et al., 2014; Appendix 3). We estimated the significance of each run against 100 null matrices obtained with two null models: the Erdös-Rényi (ER) model (Marquitti et al., 2014) and one proposed by Bascompte et al. (2003). We also ran a modularity analysis considering both habitats together. If the habitats functioned as separated units, separate modules corresponding with each community would be generated.

To evaluate the statistical significance of the estimated network metrics, we compared the observed values to 1,000 random values calculated from the null matrices. These matrices were generated using a randomization algorithm that conserves the total number of interactions per row and column in the matrix (Patefield’s r2dtable algorithm). Such a null model is not prone to type I errors (Dormann, Gruber & Fründ, 2008). The network indices (connectance, H2’, NODF, wNODF and Q) were expressed as z-scores (observed – mean(null)/sd(null)), and the statistical significance was assessed by Z-tests. The interaction networks and networks metrics were built and estimated using the bipartite package version 2.11 (Dormann, Gruber & Fründ, 2008) in R software (R Development Core Team, 2018).

Analysis of phenotypic floral integration

To obtain a measurable estimate of the magnitude (i.e., degree to which the traits are tied) and pattern (i.e., arrangement of the relationships among traits) of covariation among sets of functionally related floral traits, we estimated the phenotypic integration index (PINT). We also expressed the PINT as a percentage depending on the maximum possible integration levels (RelPINT). PINT and RelPINT were estimated using the package PHENIX (Torices & Muñoz-Pajares, 2015) in R software (R Development Core Team, 2018); both are based on a correlation matrix following Wagner (1984). We calculated the PINT for each plant species (except those lacking sufficient data) and the average PINT of plants in both communities. Since flowers with floral traits of a similar size, i.e., corolla, stamen, and style length, produce high PINT values simply by correlation, we included nectar metrics as floral traits. The reward traits were added to mitigate high PINT values unrelated with floral specialization. We obtained the average PINT across the species of each module of the overall interaction network to link phenotypic floral integration patterns and ecological specialization and assessed differences across these species with one-way ANOVAs. We also compared the average PINT across habitats following the same procedure.

Results

Hummingbirds and their floral resources

The plant-hummingbird data set comprised a total of 3,403 interactions between 26 plant species belonging to eight families and four hummingbird species. In the rainforest habitat, we recorded 1,069 interactions with 18 plant species belonging to eight families (Acanthaceae, Bromeliaceae, Costaceae, Fabaceae, Heliconiaceae, Malvaceae, Marantaceae, Rubiaceae) (Figs. 1 and 2). In the savanna habitat, we recorded 953 interactions with eight plant species belonging to two families (Bromeliaceae and Rubiaceae) (Figs. 1 and 2). The hummingbird assemblage was the same in both communities and composed of year-round species, including two species in the Emeralds clade (Chlorestes candida and Amazilia tzacatl) and two in the Hermits clade (Phaethornis longirostris and Phaethornis striigularis). In the rainforest habitat, we recorded three additional species: Anthracothorax prevostii, Heliothryx barroti and Phaeochroa cuvierii. However, they only made illegitimate visits, acting as nectar robbers. For this reason, these species were not included in the mutualistic network described below. The plant assemblages were distinct in each community, with no shared species, yet the hummingbird species were the same. Thus, both habitats are considered as a single interaction network interconnected by the hummingbirds.

Figure 1 Networks of hummingbirds and their nectar plants with identified modules indicated by colors.

(A) Ecological network comprised by the plant and the hummingbird assemblages from the two habitats, identifying modules for plants and pollinators. (B) Ecological network obtained from the rainforest habitat. (C) Ecological network from the savanna habitat. The thickness of the lines is proportional to the strength of the interactions. Circles from A to Z represent the plant species: (A) Aechmea tillandsioides, (B) Aechmea bracteata, (C) Androlepis skinneri, (D) Billbergia viridiflora, (E) Bromelia pinguin, (F) Calathea lutea, (G) Catopsis berteroniana, (H) Costus pictus, (I) Costus scaber, (J) Erythrina folkersii, (K) Heliconia aurantiaca, (L) Heliconia collinsiana, (M) Heliconia latispatha, (N) Heliconia librata, (O) Heliconia wagneriana, (P) Justicia aurea, (Q) Malvaviscus arboreus, (R) Odontonema callistachyum, (S) Odontonema tubaeforme, (T) Palicourea triphylla, (U) Psychotria poeppigiana, (V) Stromanthe macrochlamys, (W) Tillandsia bulbosa, (X) Tillandsia pruinosa, (Y) Tillandsia streptophylla, (Z) Vriesea heliconioides. Circles H1–2 represent the hummingbird Hermits clade: Phaethornis longirostris (H1) and Phaethornis striigularis (H2). Circles E1–2 represent the hummingbird Emeralds clade: Chlorestes candida (E1) and Amazilia tzacatl (E2).

Figure 2 (A) Rainforest and (B) savanna habitats with some examples of understory plant species visited by hummingbirds (C–J) photographed in the study site.

Plant species correspond to: (C) Justicia aurea (Acanthaceae), (D) Heliconia wagneriana (Heliconiaceae), (E) Bromelia pinguin (Bromeliaceae) and (F) Costus scaber (Costaceae) from the rainforest assemblage and (G) Palicourea triphylla (Rubiaceae), (H) Tillandsia pruinosa (Bromeliaceae), (I) Psychotria poeppigiana (Rubiaceae) and (J) Androlepis skinneri (Bromeliaceae) from the savanna assemblage. Photo credit: Jaume Izquierdo-Palma.

Despite we did not sample other plant-pollinator interactions, insects as floral visitors were rare in general and acted mainly as illegitimate visitors based on our observations. Most of the plant assemblage exhibited a clear ornithophilic syndrome (acting by itself as an insect deterrent) but some plant species showed less strict floral barriers against generalized pollinators, such as Catopsis berteroniana. From the video recordings we identified legitimate visits by insects in Calathea lutea and Stromanthe macrochlamys, both from the Marantaceae family. Despite Phaethornis striigularis was the only hummingbird visitor registered in both of these plant species, we observed insects (butterflies, bees and hoverflies) visiting and even opening the flowers using their complex explosive pollination mechanism (Ley & Claßen-Bockhoff, 2009). We also noticed abundant legitimate visits by different butterfly species in Psychotria poeppigiana. Thus, hummingbirds acted as occasional visitors to those species. In addition, we observed some visits in Costus pictus by euglossine bees, but we could not determine whether they were legitimate or illegitimate. In some plant species, such as Heliconia librata and Heliconia aurantiaca, we identified small butterflies, ants and small bees (mainly stingless bees of the genus Trigona) obtaining nectar from the bracts, acting as nectar robbers, but none of these observations were considered in this study.

Plant-hummingbird interaction networks

The complete network had low levels of connectance (0.35, z-score = –2.82. p = 0.005) and high levels of H′2 compared to the null matrices (0.87, z-score = 15.97, p = <0.0001), showing ecological specialization between hummingbirds and plants. The values of NODF (38.33, z-score = –3.960, p = 0.002) and wNODF (11.41, z-score = –3.89, p = <0.001) were statistically significant, showing lower levels of nestedness than expected. The Q value (0.51, z-score = 16.43, p < 0.001) indicated significant modularity that was higher than expected. We obtained three modules: one formed by Phaethornis longirostris, another by Phaethornis striigularis, and a final formed by the two Emerald species, Chlorestes candida and Amazilia tzacatl. The modules did not separate the habitats, but the habitats were related with the ecological specialization of species. The module formed by P. longirostris only included plants species from the rainforest assemblage (Fig. 3; Table S1). As we obtained the same modules using the QuaBiMo and MODULAR software, we only used the results from QuaBiMo because our network was quantitative.

Figure 3 Average floral integration index (PINT) found in each module.

Colors indicate the modules found in the complete network including the rainforest and savanna assemblages. One module is formed by Amazilia tzacatl (A. tzacatl) and Chlorestes candida (Ch. candida), one by Phaethornis striigularis (P. striigularis) and one by Phaethornis longirostris (P. longirostris). Hummingbird illustration credit: Marco Antonio Pineda Maldonado/Banco de Imágenes CONABIO.

The Hermits were the main clade of floral visitors. Phaethornis longirostris visited 16 plant species and was the only hummingbird species recorded in 11 of these, all belonging to the rainforest assemblage (Figs. 1 and 2). The strength of the interaction (represented by the number of visits/h) between P. longirostris and Heliconia wagneriana is remarkable, with a mean of 47.44 visits/h, far above any other interaction. This visitation rate can be explained by the fact that H. wagneriana grows in large patches and P. longirostris is the only hummingbird capable of obtaining nectar from their long, curved flowers (Fig. 4). Thus, they remained near the H. wagneriana patches during the flowering period, taking advantage of their abundance despite being considered trapliners. Phaethornis striigularis visited the flowers of 15 plant species and was the only visitor recorded to eight of them, five of which were in the rainforest and two in the savanna assemblage (Figs. 1 and 2). Amazilia tzacatl and Chlorestes candida visited seven and six plant species, respectively. These Emeralds always acted as generalist foragers of generalist plants species in both habitats. This is probably due to the trait mismatch between their bills and the specialized corollas, so they were not the only visitors recorded to any of the flowering plants.

Figure 4 Trait matching between corresponding pairs of morphological traits in three plant species and their exclusive hummingbird visitor, Phaethornis longirostris, in the study area.

Plant species correspond to: (A) Billbergia viridiflora (Bromeliaceae), (B) Heliconia aurantiaca (Heliconiaceae), and (C) Heliconia wagneriana (Heliconiaceae). Photo credit: Jaume Izquierdo-Palma.

When dividing the complete network by habitat, each community differed considerably in its network topography. The plant-hummingbird interaction network in the rainforest habitat had a low level of connectance (0.35, z-score = –2.88, p = 0.004) and high level of H′2 (0.83, z-score = 4.95, p < 0.0001), similar to the complete network. The NODF (25.77, z-score = –0.18, p = 0.86) was not statistically significant, yet the wNODF (10.15, z-score = –2.32, p = 0.02) was lower than expected. In the rainforest habitat, Phaethornis longirostris was the main floral visitor to 13 plant species followed by Phaethornis striigularis, which visited seven species (Fig. 1). On the other hand, Amazilia tzacatl and Chlorestes candida only visited three or two plant species, respectively. The plant-hummingbird interaction network in the savanna habitat showed higher levels of connectance compared to the rainforest community, although these were lower than expected (0.59, z-score = –2.64, p = 0.009). The H′2 value (0.47, z-score = 3, p = 0.003) was intermediate, suggesting less niche specialization than in the rainforest community (Fig. 1). The NODF (70.59, z-score = 0.71, p = 0.47) and wNODF (41.91, z-score = 1.37, p = 0.17) were higher but not significantly higher as compared to the rainforest network. In the savanna habitat, P. striigularis was the main visitor to eight plant species. On the other hand, we only recorded a few visits of P. longirostris to three plant species, with a visitation rate of 0.02 to 0.10 visits/h. For this reason, the latter species can be considered a rare visitor to the savanna habitat. Unlike the rainforest interaction network, the two Emerald species had a greater role as floral visitors and behaved as territorial in the savanna habitat, as indicated by the strength of some of their floral interactions (Fig. 1).

Plant-hummingbird trait matching

From the PCA analysis, we obtained three principal components that accumulated 88.18% of the total variance (Tables 1 and 2). Floral traits selected after the correlation analysis were corolla length, corolla curvature, nectar volume, and nectar concentration (Table S2). The first component was related with straight, small-sized flowers with dilute nectar in small quantities (PC1: 50.5% of total variance). The main plant families matching with this category were Bromeliaceae (4 species), Rubiaceae (2), Acanthaceae (1), and Marantaceae (1). Fifty-three percent were from the rainforest and the remaining 47% from the savanna. Eighty percent of the plant species were visited by Phaethornis striigularis, 20% by P. longirostris, 40% by Amazilia tzacatl, and 33% by Chlorestes candida. The second factor was related with small flowers with high nectar concentration (PC2: 19.5% of total variance). Only two species belonged to this factor, Calathea lutea and Stromanthe macrochlamys, both found in the rainforest habitat. As explained above, both of these plant species were visited mainly by insects apart from Phaethornis striigularis. Lastly, the third factor was related with flowers with moderate corolla curvature and low nectar volume (PC3: 18.18% of the total variance). The associated plant species mainly belonged to the Bromeliaceae (5 species) and Acanthaceae (3) families. Seventy-five percent were from the rainforest and 25% from the savanna. In this case, the two Hermit species were the only visitors, and they visited the same number of plant species.

Table 1 Contribution of morphological and nectar variables in the PCA analysis.

Variables contribution in the PCA analysis related to floral types according to the floral measurements (corolla length and curvature) and nectar metrics (volume and concentration) from plant species visited legitimately by hummingbirds in the study area. Total variance explained: PC1 (50.5%), PC1 (19.5%) and PC3 (18.18%).

	PC1	PC2	PC3	
Corolla length (mm)	−0.724813	−0.554099	0.104947	
Curvature (degrees)	−0.732850	0.057970	0.572058	
Nectar volume (µl)	−0.710645	−0.131267	−0.617961	
Nectar concentration (°Bx)	−0.672742	0.672500	−0.083464	

Table 2 Plant species contribution in the three principal components according to their floral and nectar measures.

Plant species contribution in the PCA analysis related to floral types according to their floral measures and nectar metrics. Family is indicated for each plant species.

	PC1	PC2	PC3	Family	
Justicia aurea	−0.20912	−1.38380	0.63834	Acanthaceae	
Odontonema callistachyum	1.54261	−0.33942	0.78373	Acanthaceae	
Odontonema tubaeforme	0.26520	0.01332	1.74687	Acanthaceae	
Aechmea tillandsioides	1.05441	0.41421	−0.49576	Bromeliaceae	
Billbergia viridiflora	−1.32573	0.53509	0.88704	Bromeliaceae	
Vriesea heliconioides	0.35424	0.19862	0.72610	Bromeliaceae	
Costus pictus	−2.96626	0.26148	−0.55565	Costaceae	
Costus scaber	−1.85085	−0.13146	0.73183	Costaceae	
Erythrina folkersii	0.36308	−2.31230	−0.23929	Fabaceae	
Heliconia aurantiaca	−1.53691	−0.74214	0.29206	Heliconiaceae	
Heliconia collinsiana	−2.58638	−0.00323	−0.54788	Heliconiaceae	
Heliconia latispatha	−1.44918	−0.29543	−2.40521	Heliconiaceae	
Heliconia librata	0.41088	0.55341	−0.39214	Heliconiaceae	
Heliconia wagneriana	−1.02959	−0.66353	0.50188	Heliconiaceae	
Malvaviscus arboreus	0.80604	−0.78644	−0.57486	Malvaceae	
Calathea lutea	−1.40318	2.44192	0.32448	Marantaceae	
Stromanthe macrochlamys	1.56472	1.00354	−0.29824	Marantaceae	
Aechmea bracteata	1.89721	0.36199	−0.16328	Bromeliaceae	
Androlepis skinneri	1.07203	0.24615	−1.24655	Bromeliaceae	
Catopsis berteroniana	1.68092	−0.31029	−0.72595	Bromeliaceae	
Tillandsia streptophylla	0.34628	−0.13030	0.57020	Bromeliaceae	
Tillandsia bulbosa	0.43713	−0.06154	0.69774	Bromeliaceae	
Palicourea triphylla	1.27467	0.39677	0.13276	Rubiaceae	
Psychotria poeppiginiana	1.28778	0.73338	−0.38821	Rubiaceae	

We observed trait matching between plants (floral traits) and hummingbirds (bill morphology) mainly in species with specialized interactions (Fig. 4; Table 3). Plant species exclusively visited by Phaethornis longirostris were differentiated by their long and curved corollas (i.e., Heliconia species). The average bill length of P. longirostris was 53.3 ± 2.88 mm (n = 30), practically identical to the average corolla length of the flowers they exclusively visited, allowing them access the nectar. This Hermit species had the most curved bill in the study area (31.82° ± 4.33, n = 18), and the flowers it visited also had higher curvature in their corollas. However, its bill was approximately three times more curved than the corolla of its visited flowers. In addition, this Hermit species was the largest hummingbird of the assemblage, with an average body weight of 5.50 g (±0.83, n = 15), which seems related with the highest average nectar volume and sugar concentration of its visited plant species. We also obtained trait matching between Phaethornis striigularis and the flowers they exclusively visited, mainly small- to medium-sized flowers with some degree of curvature (higher than 5°). The average bill length of P. striigularis was 27.39 mm (±1.23, n = 20), close to the average corolla length of its visited flowers. The average bill curvature was 25.21° (±3.40, n = 14) although, as observed with the other Hermit species, the bill curvature was higher than the average corolla curvature. Phaethornis striigularis was the smallest hummingbird of the assemblage, with an average body weight of 2.61 g (±1.32, n = 11). The average nectar volume of the flowers they exclusively visited was approximately 4.5 times lower than those visited by P. longirostris. However, the sugar concentration remained similar. Finally, small- to medium-sized flowers with less than 5° of corolla curvature were visited by several hummingbird species, mainly by the Emerald species and P. striigularis.

Table 3 Hummingbird species (or groups) associated with the average floral traits across plant species they visited legitimately.

Average floral measures and nectar metrics across plant species visited exclusively by Phaethornis longirostris, Phaethornis striigularis and visited by multiple species (Amazilia tzacatl, Chlorestes candida, Phaethornis longirostris and/or Phaethornis striigularis).

Species	Corolla lenght (mm)	Stamen length (mm)	Style length (mm)	Curvature (degrees)	Nectar volume (µl)	Nectar concentration (°Bx)	
Phaethornis longirostris	53.45 ± 15.96 (n = 11)	57.46 ± 7.15 (n = 10)	56.09 ± 6.92 (n = 10)	13.45 ± 6.75 (n = 9)	26.75 ± 16.74 (n = 10)	24.19 ± 4.73 (n = 10)	
Phaethornis striigularis	22.79 ± 11.49 (n = 8)	26.48 ± 14.92 (n = 8)	26.42 ± 16.35 (n = 7)	9.24 ± 5.93 (n = 8)	5.98 ± 5.16 (n = 7)	23.54 ± 6.30 (n = 7)	
Amazilia tzacatl, Chlorestes candida, Phaethornis longirostris, Phaethornis striigularis	18.30 ± 9.28 (n = 7)	17.56 ± 11.13 (n = 7)	20.04 ± 10.98 (n = 6)	1.62 ± 2.77 (n = 7)	24.06 ± 24.68 (n = 7)	22.55 ± 2.47 (n = 7)	

Seven out of 26 plant species received visits by two or more hummingbird species. In these seven species the average corolla length was shorter than the bill length of the two Emerald species, which was 23.52 mm (±1.53, n = 28) in Chlorestes candida and 27.9 mm (±1.97, n = 30) in Amazilia tzacatl, similar to the average bill length of P. striigularis. The main difference of the Emerald species was related to the bill curvature, with these species having the straighter bills of the assemblage, or an average bill curvature of 17.84° (±3.13, n = 20) for Ch. candida and 16.75° (±2.93, n = 13) for A. tzacatl. Correspondingly, the flowers they visited were straight or had little curvature in their corollas. Regarding body weight, the two Emerald species had intermediate values between the two Phaethornis species, or 3.35 g (±0.45 g, n = 12) for Ch. candida and 4.93 g (±0.96 g, n = 10) for A. tzacatl. The average nectar volume of the flowers they visited was similar to that of the flowers visited by P. longirostris, although the average sugar concentration was lower, corresponding with 22.55°Bx (±2.47, n = 123) (Table 3).

Phenotypic floral integration

We obtained phenotypic integration values for 22 out of the 26 plant species (Fig. 3; Table S1). The average floral integration of the plant assemblage in our study site (with nectar variables) was 19.38%, around the average (21.5%) for the angiosperms examined by Ordano et al. (2008). Sufficient data were not available for the following plant species: Aechmea tillandsioides, Billbergia viridiflora, Bromelia pinguin and Tillandsia pruinosa. In comparing the average PINT of the plant assemblages of each habitat, the savanna community had a slightly higher value (rainforest = 0.80, n = 15; savanna = 0.92, n = 7), but it was not statistically significant (F = 0.35, df = 1, p = 0.56). The results of the PINT analysis across modules suggest that specialized modules had higher values, even though they were statistically similar. The plant species integrated to the Phaethornis longirostris module had higher values (PINT = 0.92, RelPINT = 22.79%, n = 9), followed by those integrated to the Phaethornis striigularis module (PINT = 0.82, RelPINT = 20.50%, n = 8). Meanwhile, the module integrated by the two Emerald species had lower values (PINT = 0.74, RelPINT = 20.36%, n = 5), although the results from the ANOVA test showed that these differences were not significant (F = 0.25, df = 2, p = 0.79). Therefore, the relationship between the ecological specialization of modules and their phenotypic floral integration index was unclear according to these data (Fig. 3).

Discussion

As expected, we found that forbidden links and trait matching promote modularity in the plant-hummingbird system of the Lacandona rainforest. However, the low number of modules and small pollination networks likely affected the non-significant relationship between ecological specialization and phenotypic floral integration.

Our results suggest that the adjacent habitats, interconnected by the same hummingbird species, did not function as separate units but instead form a single plant-hummingbird interaction network. Thus, it is possible that two plants from the rainforest and savanna are more intimately linked through their shared hummingbird species than two plants from the same habitat with different hummingbird pollinators. According to Bergamo et al. (2017), the overlap of pollinators can influence the visitation patterns and potentially lead to indirect interactions (e.g., facilitation or competition), especially with plants with a similar floral phenotype. Nevertheless, we found that strong habitat differences in plant composition might impact some of the structural parameters when analyzed separately. The rainforest habitat was characterized by plant species with long and curved corollas, whereas small- and medium-sized flowers with straight corollas or with a little curvature (even non-ornithophilous) characterized the savanna habitat (Arizmendi & Ornelas, 1990; Araújo, Sazima & Oliveira, 2013; Maruyama et al., 2013). The lack of flowers with long corollas is probably the cause of the almost complete absence of Phaethornis longirostris in the savanna habitat. Floral morphology has been shown to play an important role in tropical hummingbird-pollination systems, influencing not only the visitors but also the strength of their interactions. For example, in plant-hummingbird interactions on the West Indies, most specialized hummingbird-pollinated plants were found in highlands and were mainly pollinated by large, long-billed hummingbirds, whereas highly generalist plants were found in dry and warm lowlands and were pollinated by small, short-billed hummingbirds in addition to insect species (Dalsgaard et al., 2009). In another case study in Brazil, Maruyama et al. (2014) highlighted the importance of traits as determinants of interaction frequencies and associated them with morphological specialization and habitat occupancy, the main network structurers, in a Neotropical savanna/forest network.

In relation to network metrics, our results showed low levels of connectance and high complementary specialization in accordance with other mutualistic networks in tropical forests (Vizentin-Bugoni, Maruyama & Sazima, 2014; Maglianesi et al., 2015; Araujo et al., 2018). The relationship between plants and hummingbirds resulted highly asymmetric: Many plants only received visits from a single hummingbird species, whereas some hummingbirds visited more than ten plant species. However, reciprocally specialized interactions are rare in nature, even in networks considered specialized (Joppa et al., 2009). Despite the low number of hummingbird species in the habitats sampled, they showed high variation in their morphological traits such as body size, bill length and foraging behavior. Morphological and behavioral differences among species enabled them to be classified into three roles in the organization of the community: Phaethornis longirostris is a high-reward trapliner and P. striigularis is a low-reward trapliner (frequently acting as a nectar robber when it is unable to access the nectar reward). And, depending on the patch quality, Chlorestes candida and Amazilia tzacatl act as territorial and generalist species (Feinsinger & Colwell, 1978). Therefore, in our network, both trait matching and forbidden links could be playing a major role in niche partitioning, shaping the network structure (Dalsgaard et al., 2011). Morphological resemblance has been found to allow the exclusive access of some species (e.g., Phaethornis species) to the most specialized flowers (Bergamo et al., 2018; Sonne et al., 2019; Sonne et al., 2020). Moreover, forbidden links regulated the interactions of the two Emerald species with less specialized bill morphologies that were unable to access flowers with long and curved corollas. Thus, variation in feeding strategies and degrees of specialization with respect to specific floral resources might be crucial for the coexistence of hummingbird species (Abrahamczyk & Kessler, 2015; Maglianesi et al., 2015; Rodríguez-Flores et al., 2019; Sonne et al., 2020).

We found that modularity was not significantly related with habitat occupancy but rather with morphological specialization (Maruyama et al., 2014). Interestingly, both Hermit species formed modules integrated by only one species. Differences in bill length and curvature may promote specialization in specific floral morphologies, as reported by Rodríguez-Flores & Stiles (2005) for the Colombian Amazon. The Hermits clade is considered the most specialized hummingbird group in regard to food resources and is highly diverse in the rainforests of South America (Rodríguez-Flores & Stiles, 2005). Given that several plant species were visited exclusively by P. longirostris and P. striigularis, these hummingbirds could be acting as “key” species for the maintenance of the plant community (De Araújo, Hoffmann & Sazima, 2018). Hermits have been previously reported to play this role in other studies and to interact with more plants than other hummingbird species, for example, Phaethornis eurynome in the Atlantic rainforest (Vizentin-Bugoni, Maruyama & Sazima, 2014) and P. petrei in the Neotropical savanna of Brazil (Maruyama et al., 2014; De Araújo, Hoffmann & Sazima, 2018). The two Emerald species, unlike the Hermits, were also observed feeding in canopy trees, always on non-specialized flowers where nectar is easily accessible (e.g., Inga vera in Fabaceae and Quararibea funebris in Bombacaceae). Additionally, in the understory, the role of the two Emerald species was more important for plant species with less restrictive morphological floral barriers, where they usually behave as territorial. Thus, in the absence of morphological specialization, the dominance hierarchy, which is correlated with body size, might play an important role in the Emeralds’ niche portioning (Rodríguez-Flores & Arizmendi, 2016, López-Segoviano, Bribiesca & Arizmendi, 2018; Márquez-Luna et al., 2019).

Contrary to expectations, we did not find higher floral integration in specialized modules or differences between habitats. Some studies have reported the absence of evidence for pollinator-mediated selection on correlated traits (e.g., Conner, 2002; Herrera et al., 2002; Meng et al., 2008). In both studied habitats, Phaethornis longirostris and P. striigularis were the only visitors to many plant species, which was consequently reflected in the floral integration. Plant species with specialized pollination systems should experience stronger or more consistent stabilizing or directional selection on floral traits than species with generalized pollination (Rosas-Guerrero et al., 2011). However, high covariation among floral and vegetative traits could be the default situation (Armbruster et al., 1999).

In this context, pollination by various functional groups would decrease the homogeneity of the pollination function and, as a result, the correlational selection on relevant floral characters and nectar properties (Berg, 1960; Ordano et al., 2008). For this reason, the fact of not having considered other invertebrate visitors should not alter our hypothesis, since specialized modules are less likely to be visited by other pollinator taxa. Some studies such as Rosas-Guerrero et al. (2011) on Ipomoea and Pérez, Arroyo & Medel (2007) on Schizanthus support the idea that floral integration in pollinator-dependent species is shaped by pollinator-mediated selection and is stronger in specialized relationships. However, these studies tested differences between different functional groups of pollinators or morphospecies, for example, between bird pollination and insect pollination systems. Thus, differences can be higher across plant species with different pollination systems when comparisons are conducted among species from the same family. Nevertheless, all plants included in both assemblages received legitimate visits from hummingbirds despite differences in their floral specialization.

Studies on plant-pollinator mutualistic networks have provided important information for understanding the underlying processes that structure communities. However, the impact of pollinators on their nutritional plants, especially those with a greater degree of specificity, has received little attention. Herein, we used a new approach with the aim of linking the underlying network structuring processes with the consequent modularity related to ecological specialization and the consequences for phenotypic floral integration in two adjacent habitats in the Lacandona rainforest of Mexico. However, we noted a main limitation in our study, which is the low number of modules and pollination networks that can limited the assessed relationships between ecological specialization of modules and their phenotypic floral integration. To assess whether specialized modules are characterized by higher phenotypic floral integration one would need larger datasets of both the number of modules and pollination networks. Thus, generalizations about the processes involved in the modular organization of plant-hummingbird networks in terms of morphological specialization (i.e., phenotypic floral integration could be made). Therefore, future work reviewing plant-hummingbird interactions and levels of flower integration should include datasets from a wider region to obtain both a higher number of specialized or generalist modules and networks of various sizes. Although the number of modules and network size were the main limitations in our study to properly assess the relationships between ecological specialization of modules and their phenotypic floral integration, it is also possible that the establishment of modules failed because the interaction network was built based on the legitimate visitation rates of their floral visitors (potential pollinators) instead of the pollinating efficiency of each hummingbird species. Therefore, we also suggest pollination experiments to determine the effectiveness and relative role of each visitor as pollinators and the breeding system of the plants species in the mutualistic networks as alternatives to determine the number and specialization levels of their modules. Clearly, this is a critical question for future research to address.

Conclusions

Mutualistic networks vary in their number of connections and the strength of interactions among species with distinct ecological specializations. Herein, we found that the plant composition of two adjacent habitats in the Lacandona rainforest may impact some of the structural parameters of the studied hummingbird-plant networks. Although the plant assemblages were distinct, the two habitats were highly interconnected by the hummingbirds, meaning that they formed a single interaction network. Forbidden links and trait matching were important mechanisms shaping the network topology, and they varied their relative importance according to the specialization of the species involved and habitat sampled. Modularity was associated with morphological specialization and, indirectly, with the habitat affinity of species. However, the low number of modules and pollination networks in our study limited our assessment of how ecological specialization affects the phenotypic floral integration among modules. Future research should seek to include datasets from a wider region to obtain a higher number of specialized or generalist modules and networks of various sizes in order to delineate further the relationship between plant-hummingbird interactions and their levels of floral integration.

Supplemental Information

Supplemental Information 1 Floral and nectar measurements.

Columns indicate the plant species, family and the different morphological/nectar variables measured. Rows represent individual measurements.

Click here for additional data file.

Supplemental Information 2 Hummingbird’s bill and weight measurements.

Columns indicate the hummingbird species and the different variables measured. Each row represents individual measurements.

Click here for additional data file.

Supplemental Information 3 R scripts.

R scripts to create the bipartite networks, estimate the network metrics and modularity, estimate the z-scores and the floral integration index.

Click here for additional data file.

Supplemental Information 4 Dataset used in the R scripts.

This folder contains the files necessary to create the bipartite matrix (rainforest, savanna and complete network), estimate the different network metrics and estimate the phenotypic floral integration mentioned in the R scripts.

Click here for additional data file.

Supplemental Information 5 Association of the modules detected by the QuanBiMo algorithm in the total network (rainforest and savanna assemblages) with the average PINT of their plant species and the average PINT of each module.

Each plant species is organized according to the module to which it belongs: Chlorestes candida (Ch. candida) and Amazilia tzacatl (A. tzacatl) module, Phaethornis longirostris (P. longirostris) module and Phaethornis striigularis (P. striigularis) module. The phenotypic integration index (PINT), the percentage of maximum possible integration (RelPINT) and the number of observations used (N) is reported for each plant species and module average.

Click here for additional data file.

Supplemental Information 6 Floral traits of plant species included in the complete mutualistic network.

For each plant species of both habitats, floral measures, nectar metrics, family identity and legitimate hummingbird visitors are shown.

Click here for additional data file.

We thank Natura Mexicana A.C., especially Javier de la Maza and Julia Carabias for the logistical support and allowing access to their facilities. We thank the staff of the Estación Biológica Chajul for trail maintenance, security, and accommodation. We thank Stuart Pimm, Pietro Kiyoshi Maruyama, and one anonymous reviewer who provided constructive comments that improved the manuscript. JI-P thanks Sergio Izquierdo and Fulvio Eccardi for being a constant source of inspiration as well as Posgrado en Ciencias Biológicas (Universidad Nacional Autónoma de México) and Universidad Autónoma de Tlaxcala for the logistical support, fieldwork equipment, and supplies. This work constitutes the partial fulfillment of JI-P’s doctorate at the Universidad Nacional Autónoma de México. This paper is dedicated to the memory of Ricardo Frias, who suddenly passed in April 2020. He was a tireless defender of the Lacandona rainforest and considerably improved this manuscript with his invaluable comments and ideas.

Additional Information and Declarations

Competing Interests

Author Contributions

Data Availability

The authors declare that they have no competing interests.

Jaume Izquierdo-Palma conceived and designed the experiments, performed the experiments, analyzed the data, prepared figures and/or tables, authored or reviewed drafts of the paper, and approved the final draft.

Maria del Coro Arizmendi conceived and designed the experiments, prepared figures and/or tables, authored or reviewed drafts of the paper, and approved the final draft.

Carlos Lara conceived and designed the experiments, prepared figures and/or tables, authored or reviewed drafts of the paper, and approved the final draft.

Juan Francisco Ornelas conceived and designed the experiments, prepared figures and/or tables, authored or reviewed drafts of the paper, and approved the final draft.

The following information was supplied regarding data availability:

Raw data and R scripts are available in the Supplemental Files.

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
