# Peer review of "Forbidden links, trait matching and modularity in plant-hummingbird networks: Are specialized modules characterized by higher phenotypic floral integration?"

_PeerJ, doi:10.7717/peerj.10974_

## Round 0.1 · original submission · Major Revisions

Dear colleagues:

I am acting as the sole reviewer of your paper. I know hummingbirds well and I have written extensively about networks, including those of pollinators. There is much I like about the paper. Particularly interesting is that you have the same set of four species in two very different habitats. The species themselves include two hermits and two emeralds. As such they have different feeding preferences.

If you will take care of my immediate concerns, I will send your revision to at least two other reviewers. Here’s my concern.

In addition to describing which species of hummingbird feed on which flowers, you are trying to draw inferences that the patterns you observe are special in some way. You want to assert that competition shapes these patterns and that is notoriously difficult to do. That different species feed on different resources may be competition, but it may simply be that it’s a consequence of them being different species.

The really problem comes when on tries to draw inferences about networks. One needs a null hypothesis for that and I don’t think you’ve got the right one. If you have it's not clear to me how you've simulated the possibilities under the null. Your number has to constrain both the row and column sums of your matrix.

Now, the best source of this discussion is my book with Jim Sanderson, Patterns of Nature (University of Chicago Press.). Asking you to buy my book is self-serving, of course. But the papers I quote below contain the main ideas as they apply to binary matrices. However, you proceed, I do need to see simulations that confirm your assertions that the feeding patterns are statistically special.

Joppa, L.N, J, Bascompte, J. M. Montoya, R.V. Sole, J. Sanderson, and S. L. Pimm. 2009. Reciprocal specialization on ecological networks. Ecology Letters 12: 961-969.

Joppa, L. N., J. M. Montoya, R. Solé, J. Sanderson, and S. L. Pimm. (2010). On nestedness in ecological networks. Evolutionary Ecology Research 12: 35-46.
Sanderson, J. G. and S. L. Pimm 2015. Patterns in Nature: the Analysis of Species Co-occurrences. University of Chicago Press, Chicago IL.

---

## Round 0.2 · Major Revisions

As you can see, the two reviewers have substantial concerns about your paper. Yes, you should resubmit this, but you will need to do a substantial amount of work to make it acceptable. It will need to be reviewed again.

Reviewer 1 ·

Basic reporting

The manuscript requires careful and extensive English language editing throughout. In its current state, the manuscript is difficult to follow and has no clear direction. The term "phenotypic floral integration" is used frequently, but is not explained. Many paragraphs are too long and have no clear objective.

Experimental design

While the methods used may be appropriate, I struggle to follow the manuscript in its current state.

Validity of the findings

Again, I cannot comment on the validity of the findings given the above.

Additional comments

I recommend that the authors either invite a native English speaker on as a co-author, or use an English editing service (these can be extremely helpful). The authors have conducted some excellent field work and made valid observations. However, it is unfortunate that these interesting observations are lost in poorly organized paragraphs and improper sentences. Consequently, I cannot recommend this manuscript for publication in PeerJ in its current state. Below is a list of comments I made before deciding to recommend reject:

L25-26: "allow the niche portioning and ultimately, modularity.". I don't understand what the authors are trying to say here. I believe they are referring to niche partitioning, which is the term used later in the manuscript. Please go through the manuscript and remove all excessive and particles such as "the" and "a/an" where they are not needed. Nouns do not always need to be associated with these articles, this is consistent difference between the Spanish and English languages.

L26-28: I don't understand this sentence. It has no clear direction. The term "floral integration" is used as if every reader is supposed to know it. This needs to be explicitly described in the introduction, and probably removed from the abstract. It should instead be described using plain English and no topic-specific jargon.

L39: "networks dynamics" should be "network dynamics"

L303-305: I'm unsure what constitutes "illegitimate" visits. Why were these Hummingbird species not included in the study? This is brushed over and not explained well.

L402: "phenotypic floral integration". Again, this term needs to be explicitly explained from the outset before it is used throughout.

L419: "sedentary species". This is probably not the term you want here. Birds are certainly not sedentary. Do you mean they do not migrate between habitats?

L429-430: remove "in this species rich tropical community".

L432: remove "in a diverse tropical community".

L437-439: I don't understand what the authors are trying to convey here. What is the point and how does it relate to what is described above (particularly nestedness)?

L452-503: this is an enormous paragraph that could easily be shortened with some careful editing. It is also poorly structured and hard to follow.

I find the authors arguments hard to follow. Does trait-matching promote specialization and modularity or the other way around?

·

Basic reporting

In this manuscript, Palmas and colleagues present an association between morphological constraints in plant-hummingbird interaction networks from two distinct habitats in Mexico.
The main idea of associating network structure of the interaction network, with emphasis in modularity, with phenotypic integration of the flower assemblages is indeed very interesting. Data collection seem to be alright and overall, this is study that brings new ideas. Good job done.
I have two major concerns though related to the study, meaning that it would require major revising. One more conceptual and one methodological/choice of the analytical framework given their sampling of two interconnected habitats:
- Morphological matching (trait complementarity) and forbidden links (understood as exploitation barrier) are not the same thing, albeit related. This needs to be properly understood by the authors, and the theoretical background of the manuscript should be thoroughly revised (see Santamaría & Rodrígues-Gironés 2007,

Experimental design

- From the species composition and interactions in the two habitats, it is clear that the two studied habitats are highly interconnected by the hummingbirds. Although plant assemblages are distinct, this means that the two habitats form a single network of interaction. It could even be that two plants from different habitats are more intimately linked with each other through their shared hummingbird species than two plants of the same habitat with different hummingbird pollinators (see Bergamo et al. 2017, Ecology), and hence affect all consequences on floral integration that the authors are trying to show. Therefore, it makes more sense that an analysis considering one whole network and then contrasting the floral integration among modules should is conducted. At the very least, the authors need to show that these two habitat networks work as complete separate units. This may be the case as shown previously (Maruyama et al. 2014), but it seems not (modules do not reflect habitats when analyzing the whole network).
The English writing present many misspellings and inconsistencies, so a thorough revision is needed.

Validity of the findings

no general comment, see detailed comment below.

Additional comments

Detailed comments.
Abstract
Line 22. Plant-pollinator mutualistic networks, change throughout the text (also at line 62 and other places).
Line 23. “However, pairwise interactions show high variability in the number of interacting species” This sentence is strange, pairwise interactions should by definition, include two species each
Line 25. Niche partitioning.
Line 39. “The rainforest network showed higher levels of specialization and mainly composed of species of the Hermits clade…” How can this be if both habitats have the same hummingbird species?
Line 49. The conclusion presented refer to forbidden links and trait matching for one habitat, and agonistic behavior for other habitat as mechanisms structuring the interactions. But how this conclusion was reached is not clear and up to this point are it is not mentioned regarding the methods/results on what done and was found regarding this finding. Please explain and provide an abstract that more clearly present what you did and concluded based on what.
Line 52.What is “degree of specialization of pairwise interactions” To my understanding, degree and specialization are estimated for each species.
Line 84. What would be these upper and lower levels? Consumer and resources? Please clarify. What about other type of interactions? Do not we also find constraints on interactions?
Line 86-88. Trait matching and forbidden links (exploitation barrier) are two somewhat distinct concepts (e.g. Santamaría & Rodríguez-Gironés 2007; Sazatornil et al. 2016). Please check this and update it accordingly throughout.
Lines 103-107. This is not really related to your study, which is rather local scale.
Line 111. There is no idea of “opposition” as indicated by “Contrarily”.
Introduction. Maruyama et al. 2014 clearly linked modularity of a plant-hummingbird network with morphological traits and spatial distribution of species. Since your study touch upon this topic heavily, it could be explored better in Introduction to give context for your study.
Methods
Line 182-183. Why did you focus on understory plants only? Was this a limitation? Make this clear
Line 190. Were hummingbirds the main pollinators of all plant species? Make this clear.
Line 215-216. A citation for this one third addition?
Line 248. Bipartite should be in small case letters.
Line 255. It is somewhat affected, you could say “less affected” instead of “unaffected”
Line 277-282. This is a new approach used in this context, as such it needs to be better explained on what was done. At what scale was phenotypic integration calculated? For each species? For the entire assemblage (an average across species?). So that you could relate to the modules? It is important that we can understand this and that future studies can be based on yours.
Results
Line 293. “two belonged to the”
Line 296-297. “Two sub-networks with completely different plants without any shared species were clearly differentiated by habitat” How can you support this? Although plants are not, hummingbirds are the same between habitats. You should run a modularity analysis considering both habitats together since the two habitats form a single network of interaction connected by hummingbirds. It could even be that two plants from different habitats are more intimately linked with each other through their shared hummingbird species than two plants of the same habitat with different hummingbird pollinators (see Bergamo et al. 2017, Ecology), and hence affect all consequences on floral integration that the authors are trying to show. Therefore, it makes more sense that an analysis considering one whole network and then contrasting the floral integration among modules should is conducted.
Line 348-351. How was division of flowers determined? Can you support this without being arbitrary?
Line 356-357. Show the curvature of bill and corolla in the same scale
Line 360. “largest hummingbird”
Lines 369-384. Many of the info that is already present in the table, or it should be.
Line 390-392. Average across plant species? This needs to be clear.
Discussion
Lines 409. The patterns reported is also very similar with Maruyama et al. 2014.
Line 452-456. Needs to better define trait matching and forbidden links, their differences.
Line 484-488. The authors may also want to consider Maruyama et al. 2013 which specifically discuss this pattern quantitatively.
Lines 505-512. No biological process being discussed here. Suggest deleting.
Line 517. “where only plant traits differed from the modules” what do you mean?
Line 517-524. You did not analyze this in your study, so I do not think that is relevant discussion.
Line 527. “did not differed” please correct.
Lines 561-566. Too long and speculative.
Lines 569-570. Confusing sentence, please reformulate
The Discussion section is too long and could be definitely made more concise.
The Conclusion sections here seem unnecessary as the authors are only repeating their findings which are listed. The last paragraph of the Discussion already served as a conclusion (which was better written than the text in the Conclusion section).
Figure 1. Order the hummingbird species in the same order for the two networks.
Figure 3. Do you have similar images for the small flowers?
Table 1. Show this information in a figure, with the averages (boxplot? barplot?) and report the table as supplement.
References used in this review.
Bergamo et al. (2017). The potential indirect effects among plants via shared hummingbird pollinators are structured by phenotypic similarity. Ecology, 98(7), 1849-1858.
Maruyama et al. (2014). Morphological and spatio‐temporal mismatches shape a neotropical savanna plant‐hummingbird network. Biotropica, 46(6), 740-747.
Maruyama et al. (2013). Pollination syndromes ignored: importance of non-ornithophilous flowers to Neotropical savanna hummingbirds. Naturwissenschaften, 100(11), 1061-1068.
Santamaría & Rodríguez-Gironés (2007). Linkage rules for plant–pollinator networks: trait complementarity or exploitation barriers?. PLoS Biol, 5(2), e31.
Sazatornil et al. (2016). Beyond neutral and forbidden links: morphological matches and the assembly of mutualistic hawkmoth–plant networks. Journal of Animal Ecology, 85(6), 1586-1594.

---

## Round 0.3 · Major Revisions

As you can see, I'm afraid that while you have satisfied one reviewer, the other still has some substantial concerns. I must ask you to fix these and send me another revision. (S)he is concerned about whether the data you present are sufficient to address the issue of modularity.

·

Basic reporting

I consider that all my previous considerations were incorporated.

Experimental design

no comment

Validity of the findings

no comment

Additional comments

I annotated in the attached manuscript pdf, some suggestions on small edits.
Good luck.

Pietro

Reviewer 3 ·

Basic reporting

no comment (see general comments)

Experimental design

no comment (see general comments)

Validity of the findings

not comment (see general comments)

Additional comments

“Forbidden links and trait matching promote modularity in
plant-hummingbird networks: the influence of floral
integration”

The present study aims to analyse processes related to the modular organization of plant-hummingbird networks in terms of morphological specialization and their implications on the phenotypic floral integration within modules. For this, authors selected two adjacent habitats (lowland rainforest and savanna-like vegetation) in Chajul Biological Station (Mexico) to record plant-bird interactions through two consecutive years. The obtained (and contrary to expectations) results show that floral integration within specialized modules were not significant higher that other modules, non-specialized.

The topic of this MS is potentially interesting. However, some aspects of the MS (see below) need to be considered in order to improve it. My principal concern is related to the number of modules obtained (just 3) because of small studied pollination network (26 plants and 4 bird species). It’s this value (n=3) good enough to test hypotheses about specialization and floral integration among modules? Besides, other aspects need to be considered through the MS. For example:

Title: Maybe authors need to change it. Its confuse. It would be more easily if authors add, in some way, the principal conclusion of this study: “specialized modules are not characterized by higher phenotypic floral integration”

Abstract. L23: Substitute “pairwise interactions” by “among species interactions”, or “mutualistic interactions”, or “plant-pollinator interactions”.

Abstract. L26-27: If not easy to understand the next sentence: “Thus, plants belonging to specialized modules should integrate their floral traits to optimize the pollination function” if all pollinator species (and plants) within each habitat were not included (just only understory plant and hummingbird species).

Abstract. L30: Maybe it is not necessary to explain the term “phenotypic floral integration” in the Abstract.

Abstract. L49-55: Maybe include this sentence from L47-48: “morphological specialization plays a minor role in the phenotypic floral integration” in Conclusion part.

Introduction. L 58-60: These two sentences need to be change. The first one is confuse and very general. In the second one, plant-pollinator networks just imply many species.

Introduction. L117: As previous specific terms, “modular structure” need to be explained briefly here.

Introduction. I miss some clear hypothesis to be tested. In general the Introduction is confuse. Need to be more focusing on “modularity and floral integration” and clarify the use of some terms such as “niche segregation by ecological specialization”. Besides, what about to remove sentences L102-108?

Methods. Authors need to consider if including partial plant-pollinator networks to include only understory species is good enough to study evolutionary trends (i.e. phenotypic integration) (L186). Besides, what about other pollinator species? Many plant species visited by hummingbirds are also visited by insects. Authors need to discuss in some way these aspects. It’s no clear at all if some selected plant species are also legitimate visited by insects (L241). Besides, I miss some information such as number of plant species inhabiting these habitats? And how many being visited by hummingbirds? How many hummingbird species? From these total values, how many were included in the study? What about sampling efforts for each plant species? It would be possible to include flower colour (L199)? How flower curvatures were measured (L203)? To know the accumulated nectar volume, how many hours each flower was excluded to pollinators, just 24h? The bipartite matrix included quantitative data (L253)? It’s really necessary to separate both adjacent habitats (L256)? For example, in savannah only 8 plant and 4 hummingbird species were included. What about the version of the bipartite package used (L289)?

Results. L311-325. Authors need to clarify if only birds visited these plant species. What about insects such as pollinators? What about to remove results by each habitat separately (L357-376).

Authors fail to find significant relationship between ecological specialization of modules and their phenotypic floral integration. This result could be related to the low number of plant species studied and number of obtained modules (3).

---

## Round 0.4 · Major Revisions

I don't expect to send your next revision out for review. That said, I agree with the reviewer that you really don't have enough modules — three — to draw the conclusions that you would like. So, I want you to add a paragraph or two where you concede that reviewer has a good point: this work needs to be expanded and that what you have done is limited. Please take care to accommodate the reviewer's concerns — they are ones that I share. What the reviewer suggests is that you do a completely different study. (S)he is right, of course, but I don't want to reject your paper and to require such a massive extension of work would cause me to do that.

In your cover letter with the revision, be explicit in how you address these issues and the inevitable limitations of your study. If you convince me you have done that well, I will accept the paper and not send it for further review.

Reviewer 3 ·

Basic reporting

no comment

Experimental design

no comment

Validity of the findings

not comment

Additional comments

In this new version of the MS authors have already included practically all my previous comments. The manuscript has been improved considerably, and authors have also included more information about. However, at time to analyse processes related to the modular organization of plant-hummingbird networks in terms of morphological specialization (i.e. phenotypic floral integration) I think that dataset are not good enough and its related to low number of obtained modules because of small studied pollination networks (26 plants and only 4 bird species). I mean, to analyse flora integration in just 3 modules its not good enough to conclude “specialized modules are not characterized by higher phenotypic floral integration”.

Instead I’m not sure about the difficult to obtain more data, and because I like the author’s novelty idea, I encourage them e.g. to review plant-hummingbird interactions in a wide region to obtain a higher network size. After, by using previous publications or just some flower pictures, maybe authors can obtain some of biometrical traits to analyse flower integration (maybe it is not necessary to obtain values from all traits in all plant species). Then, in this way authors will have the chance to increase number of specialized or generalist modules for a proper comparisons of their flower integration level. Authors could confirm their present results by using a smaller network size in order to generalize their principal conclusion. The MS is focusing to know relationship between ecological specialization of modules and their phenotypic floral integration.

---

## Round 0.5 · accepted · Accept

Thank you for taking care of the last set of comments. As someone who does field work, I know that there are always limitations on what a single study can achieve. I think your paper explains how and why you collected the data and the methods to analyse them. In this revision, you make the study's limitations explicit and provide guidelines of how new studies could move it forward.

My best wishes and thanks for choosing PeerJ.